# Is Medical Pretraining Enough When the Modality Is Different? A Study on Endoscopic Polyp Segmentation

**Dipika Boro**[1]                                                    DIPIKA_BORO@UML.EDU
**Yu Cao**[1]                                                              YU_CAO@UML.EDU
**Benyuan Liu**[1]                                                  BENYUAN_LIU@UML.EDU
**Qilei Chen**[1]                                                    QILEI_CHEN1@UML.EDU
[1] *University of Massachusetts, Lowell.*

**Editors:** Accepted for publication at MIDL 2025

## Abstract

Using pretrained models for fine-tuning is a widely adopted strategy in medical imaging, where labeled data is scarce. While ImageNet remains the standard for pretraining in computer vision, RadImageNet, a radiology-specific alternative, has shown promise in radiology realted vision tasks. However, its effectiveness in non-radiological modalities like endoscopy remains unclear. In this study, we conduct a focused evaluation of how transfer learning from natural or medical images affects performance in endoscopic polyp segmentation, using ImageNet, RadImageNet, and a histopathology dataset for pretraining. Two backbone architectures—ResNet-50 and ViT-Small are integrated with a DeepLabV3+ decoder and evaluated on three public datasets: CVC-ClinicDB, Kvasir-SEG, and SUN-SEG. ImageNet-pretrained models consistently outperform those pretrained on medical datasets. These results highlight that medical-domain pretraining is not universally beneficial and emphasize the need for modality alignment when selecting pretrained models for medical imaging tasks. Github - https://github.com/dipikaboro2/med-pretraining

**Keywords:** Transfer learning, Pretraining, Polyp segmentation, ViT, ResNet

## 1. Introduction

Transfer learning with pretrained models is a widely adopted strategy in computer vision, offering improved generalization, faster convergence, and more efficient training. This approach is especially valuable in domains where annotated data is limited, such as medical image analysis. ImageNet (Deng et al., 2009) remains the default pretraining dataset due to its large scale and strong generalization capabilities across diverse tasks. However, ImageNet comprises natural images that differ substantially in structure and appearance from medical images.

RadImageNet (Mei et al., 2022) was proposed as a radiology-specific alternative to ImageNet, comprising of over a million medical images from CT, MRI, and ultrasound modalities. It has shown performance gains in several radiology-focused tasks. However, medical imaging spans a variety of modalities beyond those in RadImageNet, such as endoscopy and histopathology, which differ significantly in structure, color, and visual semantics. The NCT-CRC-HE-100K (Kather et al., 2018) dataset is a pathology image collection consisting of 100,000 histological images of human colorectal cancer and healthy tissues. Whether medical image pretraining, radiology-specific or histopathology-specific, generalizes to polyp segmentation in endoscopic images remains largely unexplored.

This study presents a systematic evaluation of ImageNet, RadImageNet and histopathology images as pretraining sources for endoscopic image segmentation. We investigate two representative encoder architectures—ResNet-50 (He et al., 2016), a CNN based architecture, and ViT-Small (Dosovitskiy et al., 2021), a transformer based architecture—each integrated into a unified DeepLabV3+ (Chen et al., 2018) decoder. In order to get a comprehensive analysis of performance across the pretraining domains, these models are evaluated on three publicly available polyp segmentation benchmarks: CVC-ClinicDB (Bernal et al., 2015), Kvasir-SEG (Jha et al., 2020), and SUN-SEG (Fan et al., 2020; Ji et al., 2021, 2022).

## 2. Method

### 2.1. Framework

As shown in Figure 1, we follow an encoder-decoder framework, with ResNet-50 or ViT-Small as the encoder and a DeepLabV3+ decoder to generate the segmentation output.

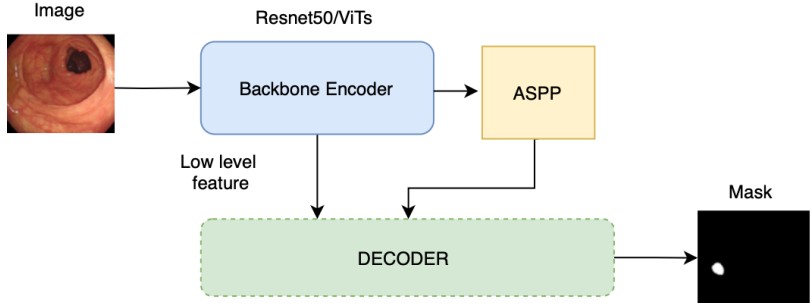

Figure 1: Architectural overview of the encoder-decoder framework.

Input images are resized to $224 \times 224$ and passed through the encoder to extract features. For ResNet-50, the final convolutional layer output is fed into an Atrous Spatial Pyramid Pooling (ASPP) module to capture multi-scale context, while a low-level feature map from an earlier layer (layer1) is used in a skip connection to the decoder. For ViT-Small, intermediate feature maps are obtained. The final transformer block output is processed by ASPP, and the earliest feature map serves as the decoder skip connection. The decoder combines ASPP output with low-level features and produces a segmentation mask.

### 2.2. Experimental Setup

We evaluate our pretrained models on three publicly available polyp segmentation datasets: CVC-ClinicDB (612 images), Kvasir-SEG (1,000 images), and SUN-SEG (49,136 images). Each dataset is split 80/20 into training and validation sets using a fixed seed for reproducibility.

We train using binary cross-entropy loss and the Adam optimizer (learning rate 0.0001) for 20 epochs with a batch size of 32 on a single NVIDIA TITAN RTX GPU. Performance is evaluated on the validation set using Dice coefficient and Intersection over Union (IoU) score. Final results are reported using the best model checkpoint.

## 3. Results and Conclusion

Table 1 presents segmentation performance across the three datasets for backbones pretrained on ImageNet, RadImageNet and NCT-CRC-HE-100K. ImageNet-pretrained models consistently outperform RadImageNet or NCT-CRC-HE-100K pretrained ones, with the largest differences on smaller datasets. This pattern holds for both CNN and transformer architectures. Sample predicted masks are shown in Figure 2.

Table 1: Dice and IoU scores of finetuned models.

| Model | Dataset | ImageNet | | RadImageNet | | Histopathology | |
|---|---|---|---|---|---|---|---|
| | | Dice | IoU | Dice | IoU | Dice | IoU |
| ResNet50 | CVC-DB | 0.8230 | 0.7389 | 0.5820 | 0.4634 | 0.7262 | 0.6131 |
| | Kvasir-SEG | 0.8244 | 0.7340 | 0.5279 | 0.4006 | 0.7062 | 0.5947 |
| | SUN-SEG | 0.9353 | 0.8862 | 0.9209 | 0.8657 | 0.9307 | 0.8813 |
| ViT-S | CVC-DB | 0.8705 | 0.7913 | 0.5033 | 0.4024 | 0.5520 | 0.4325 |
| | Kvasir-SEG | 0.8706 | 0.7986 | 0.5228 | 0.3954 | 0.5065 | 0.3839 |
| | SUN-SEG | 0.9000 | 0.8334 | 0.8411 | 0.7580 | 0.8453 | 0.7658 |

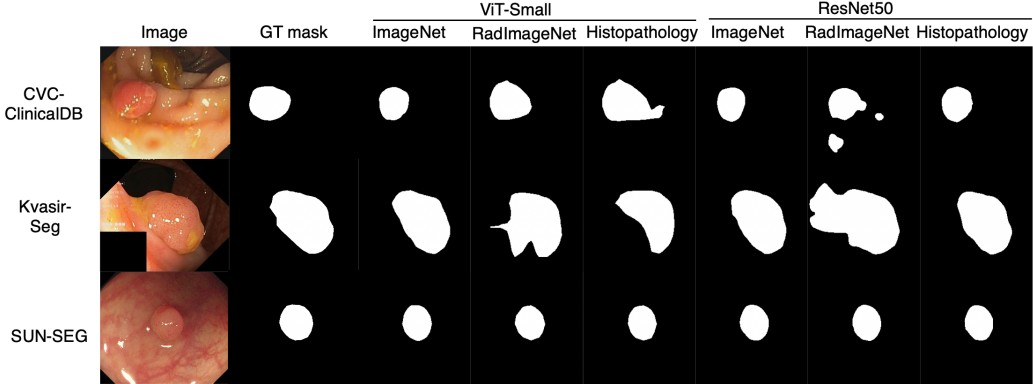

Figure 2: Samples of predicted masks from finetuned ViT-small and ResNet50 models.

The performance gap narrows on SUN-SEG, the largest dataset in our evaluation. Models pretrained on histopathology data perform comparably to those pretrained on ImageNet, with only slightly lower Dice and IoU scores, whereas RadImageNet-pretrained models continue to underperform. This may be attributed to the similarity in color characteristics between histopathology and endoscopic images, in contrast to grayscale radiology images.

These findings suggest that although larger datasets can reduce the impact of pretraining misalignment, features learned from ImageNet remain more transferable to non-radiological modalities such as endoscopy. Overall, our results indicate that medical-domain pretraining is not universally advantageous, and that modality-specific characteristics should guide the selection of pretrained models in medical imaging tasks.

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
