# OpenReview forum: "Is Medical Pretraining Enough When the Modality Is Different? A Study on Endoscopic Polyp Segmentation"
_MIDL.io/2025/Short_Papers — MIDL 2025 - Short Papers_

### Official Review · Reviewer_hRuj · 2025-04-28

**Rating:** 3
**Confidence:** 5

**Summary:**

The authors test a simple but practically relevant question: Should we pre-train on “medical” images if the downstream task is endoscopic polyp segmentation?
They fine-tune two back-bones—ResNet-50 and ViT-Small—each initialised from ImageNet or RadImageNet and evaluate them on three public polyp data sets (CVC-ClinicDB, Kvasir-SEG, SUN-SEG). In every setting the ImageNet initialisation performs better; the gap is large on the small data sets (e.g. Dice 0.82 → 0.58 on CVC-ClinicDB) and shrinks on the 50 k-image SUN-SEG set.

**Strengths:**

+ Clear, controlled comparison – only the pre-training corpus changes, making conclusions unambiguous.
+ Architecture robustness – consistent trend across both a CNN and ViT.
+ Open code & reproducibility – repository provided.

**Weaknesses:**

1. Limited novelty – “obvious” RGB vs. greyscale argument. Because natural and endoscopic images are both 3-channel RGB, while CT/MRI/US images in RadImageNet are single-channel, many practitioners would expect ImageNet to win. The study largely confirms this intuitive expectation rather than uncovering a surprising phenomenon.
2. No deeper analysis of why RadImageNet fails. The paper could strengthen its impact by:
 * colourising RadImageNet images to test if channel count alone explains the gap,
 * probing feature similarity (e.g. CKA) to pinpoint layer-wise divergence, or
 * adding another RGB-medical modality (histopathology) to see whether “medical → medical” helps when colour statistics match.
3. Absence of statistical significance tests. No confidence intervals or p-values are reported to show the robustness of performance gaps.

---

### Decision · Program_Chairs · 2025-05-01

Accept